# Benchmarking End-To-End Performance of AI-Based Chip Placement Algorithms

## Abstract

The increasing complexity of modern very-large-scale integration (VLSI) design highlights the significance of Electronic Design Automation (EDA) technologies. Chip placement is a critical step in the EDA workflow, which positions chip modules on the canvas with the goal of optimizing performance, power, and area (PPA) metrics of final chip designs. Recent advances have demonstrated the great potential of AI-based algorithms in enhancing chip placement. However, due to the lengthy workflow of chip design, the evaluations of these algorithms often focus on *intermediate surrogate metrics*, which are easy to compute but frequently reveal a substantial misalignment with the *end-to-end performance* (i.e., the final design PPA). To address this challenge, we introduce ChiPBench, which can effectively facilitate research in chip placement within the AI community. ChiPBench is a comprehensive benchmark specifically designed to evaluate the effectiveness of existing AI-based chip placement algorithms in improving final design PPA metrics. Specifically, we have gathered 20 circuits from various domains (e.g., CPU, GPU, and microcontrollers). These designs are compiled by executing the workflow from the verilog source code, which preserves necessary physical implementation kits, enabling evaluations for the placement algorithms on their impacts on the final design PPA. We executed six state-of-the-art AI-based chip placement algorithms on these designs and *plugged* the results of each single-point algorithm into the physical implementation workflow to obtain the final PPA results. Experimental results show that even if intermediate metric of a single-point algorithm is dominant, while the final PPA results are unsatisfactory. This suggests that the AI community should concentrate more on enhancing end-to-end performance rather than those intermediate surrogates. We believe that our benchmark will serve as an effective evaluation framework to bridge the gap between academia and industry.

## 1 Introduction

The exponential growth in the scale of integrated circuits (ICs), in accordance with Moore's law, has posed significant challenges to chip design (Huang et al., 2021; Lopera et al., 2021). To handle the increasing complexity, many electronic design automation (EDA) tools have been developed to assist hardware engineers. As shown in Figure 1, EDA tools automate various steps in the chip design workflow, including high-level synthesis, logic synthesis, physical design, testing and verification (Huang et al., 2021; Sánchez et al., 2023).

Chip placement is a critical step in the chip design workflow, which aims to position chip modules on the canvas, with the goal of optimizing the performance, power, and area (PPA) metrics of final chip designs (Cheng et al., 2023; Shi et al., 2023; Lin et al., 2019). Traditionally, this is done manually by human expert designers, which costs much labor and necessitates much expert prior knowledge. Therefore, a lot of design automation methods, especially those AI-based algorithms, have been developed to automate this process. These methods mainly fall into two categories: optimization-based methods and reinforcement learning (RL)-based methods (Geng et al., 2024). Optimization-based methods employ traditional optimization algorithms, such as simulated annealing (SA) (Vashisht et al., 2020) and evolutionary algorithms (EA) (Shi et al., 2023) to directly address the large-scale optimization problem, exploring the design space to identify near-optimal solutions. In recent research, macro placement has been formulated as a Markov Decision Process (MDP), where the macro positions are determined sequentially (Mirhoseini et al., 2021). Reinforcement learning (RL)

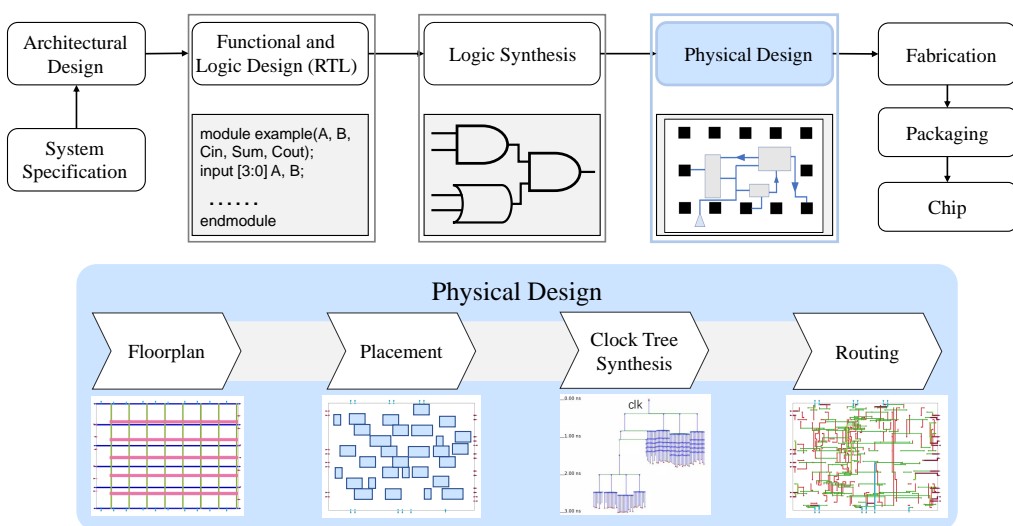

Figure 1: Illustration of the modern chip design workflow.

has emerged as a promising technique for this task due to its ability to continuously improve performance based on feedback from the environment through trial and error (Cheng & Yan, 2021; Cheng et al., 2022; Lai et al., 2022; 2023).

However, due to the lengthy workflow of chip design, the evaluations of these algorithms often focus on *intermediate surrogate metrics*, which are easy to compute but frequently reveal a substantial misalignment with the *end-to-end performance* (i.e., the final design PPA). On one hand, obtaining the end-to-end performance of a given chip placement solution requires a large amount of engineering efforts due to the lengthy workflow of chip design. In particular, we found that directly using the existing open-source EDA tools and chip placement datasets even usually fails to obtain the end-to-end performance. Thus, existing AI-based chip placement algorithms (Lai et al., 2023; Geng et al., 2024) train and evaluate learned models using intermediate surrogate metrics which are simple and easy to obtain. On the other hand, as the PPA metrics are reflected by many aspects that have not been adequately considered in the previous stages, there exhibits a critical gap between the surrogate metrics and the final PPA objectives. Therefore, this gap significantly limits the use of existing AI-based placement algorithms in practical industrial scenarios.

To address this challenge, we propose ChiPBench, a comprehensive benchmark specifically designed to evaluate the effectiveness of existing AI-based chip placement algorithms in improving final design PPA metrics. Appealing features of ChiPBench include its fully open source and reproducible characteristics, covering the entire EDA workflow from the source verilog code, and unifying the evaluation framework of AI-based chip placement methods using end-to-end performance. Thus, ChiPBench can effectively facilitate research in chip placement within the AI community by taking the first step towards a fully reproducible unified evaluation framework using the end-to-end performance. In terms of the dataset, we have gathered 20 circuits from various domains (e.g., CPU, GPU, and microcontrollers). Then, these designs are compiled by executing the workflow from the verilog source code, which preserves sufficient physical implementation kits, enabling evaluations for the placement algorithms on their impacts on the final design PPA. In terms of the evaluated algorithms, we executed *six* state-of-the-art AI-based chip placement algorithms on the aforementioned designs and *plugged* the results of each single-point algorithm into the physical implementation workflow to obtain the final PPA results.

Experimental results show that even if intermediate metric of a single-point algorithm is dominant, while the final PPA results are unsatisfactory. Moreover, visualization experiments demonstrate that intermediate metrics have weak correlation with the final design PPA, emphasizing the importance of developing algorithms towards optimizing the final design PPA rather than intermediate metrics.

Table 1: Comparison Between Our Dataset and Existing Datasets

| Dataset | Complete Design Suite | Logic Synthesis Support | Physical Design Support | Full EDA Flow Support | Large Scale &Diversity |
|---|---|---|---|---|---|
| **ISPD2005** | ✗ | ✗ | ✓ | ✗ | ✗ |
| **ICCAD2015** | ✗ | ✗ | ✓ | ✗ | ✓ |
| **EPFL Benchmarks** | ✗ | ✓ | ✗ | ✗ | ✗ |
| **OpenABC-D** | ✗ | ✓ | ✗ | ✗ | ✓ |
| **CircuitNet 2.0** | ✗ | ✗ | ✓ | ✗ | ✓ |
| **ChiPBench (Ours)** | ✓ | ✓ | ✓ | ✓ | ✓ |

This suggests that the AI community should concentrate more on enhancing end-to-end performance rather than those intermediate metrics. We believe that our benchmark will serve as an effective evaluation framework to bridge the gap between academia and industry.

We summarize our major contributions as follows. (1) Our proposed ChiPBench is a reproducible and unified evaluation framework of existing AI-based chip placement algorithms using end-to-end performance with fully open source EDA tools. This can effectively facilitate research in chip placement within the AI community. (2) We collected 20 circuits from various domains, and construct a dataset by executing the EDA workflow from the verilog source code, preserving sufficient physical implementation kits for end-to-end performance evaluation. (3) We evaluate six state-of-the-art AI-based chip placement algorithms, including most popular AI-based chip placement algorithms. (4) Experiments demonstrate that intermediate metrics have weak correlation with the final design PPA, emphasizing the critical importance of developing algorithms towards optimizing the final design PPA rather than intermediate metrics.

## 2 RELATED WORK

**Datasets** Some well-known EDA conferences, such as ISPD and ICCAD, host contests addressing EDA challenges and offer benchmarks with processed data for researchers. However, in the early years (e.g., ISPD2005 (Nam et al., 2005) and ICCAD2004 (Adya et al., 2009)), the provided datasets used overly simplified `Bookshelf` formats, which are abstracted versions of the actual design kits. Therefore, we cannot evaluate the final PPA of the placement results on those datasets. Recently, ISPD2015 (Bustany et al., 2015) and ICCAD2015 (Kim et al., 2015) have offered benchmarks and datasets closer to real-world applications, including necessary netlist, library, and design exchange files, broadening their utility slightly. Nevertheless, they still lack the essential information (e.g., necessary design kits) to run the open-source EDA tools such as OpenROAD (Kahng & Spyrou, 2021). Beyond these conferences, some other datasets have been developed in various directions. For example, the EPFL (Amarú et al., 2015) benchmarks and the larger OpenABC-D Chowdhury et al. (2021) dataset concentrated on synthetic netlists, primarily for testing modern logic optimization tools with a focus on logic synthesis. CircuitNet 2.0 (Jiang et al., 2023), on the other hand, shifted the focus towards providing multi-modal data for prediction tasks, enhancing the capability for various prediction tasks through the use of diverse data modalities. Compared with previous efforts, our proposed dataset focuses on the entire EDA workflow. As shown in Table 1, a comparison between our dataset and mainstream datasets highlights the distinctions. It provides complete files for each case and necessary design kits, such as timing constraints, library files, and LEF files, offering a comprehensive dataset that supports all stages of physical implementation and fosters a more integrated approach to chip design and evaluation.

**Placement Algorithms** Recent advancements in AI technology within the EDA field have led to a variety of AI-based chip placement algorithms. (1) Black-Box Optimization methods. Simulated Annealing (Cheng et al., 2023) provides a probabilistic method for finding a good approximation of the global optimum. Wire-Mask-Guided Black-Box Optimization (Shi et al., 2023) uses a wire-mask-guided greedy procedure to optimize macro placement efficiently. (2) Analytical methods. DREAMPlace (Lin et al., 2019) uses deep learning toolkits to achieve over a 30x speedup in placement tasks. AutoDMP (Agnesina et al., 2023) leverages DREAMPlace for the concurrent placement of macros and standard cells, enhancing macro placement quality. (3) Reinforcement Learning methods. MaskPlace (Lai et al., 2022) treats chip placement as a visual representation learning

problem, reducing wirelength and ensuring zero overlaps. ChiPFormer (Lai et al., 2023) employs offline reinforcement learning, fine-tuning on unseen chips for better efficiency. The evaluation of these algorithms mainly focuses on intermediate metrics. In contrast, we utilized the *end-to-end performance* to evaluate six existing AI-based chip placement algorithms, encompassing a significant portion of mainstream AI-based placement algorithms.

## 3 BACKGROUND ON ELECTRONIC DESIGN AUTOMATION

Electronic Design Automation (EDA) is a suite of software tools vital for designing and developing electronic systems, primarily integrated circuits (ICs). These tools enable electrical engineers to efficiently transform innovative concepts into functional products, addressing the increasing complexity and demands of modern chip design. EDA optimizes the entire design process from schematic capture to layout and fabrication, reducing time-to-market and enhancing design precision and sophistication. In the chip design workflow, EDA tools support various functions: they perform simulations to verify circuit behavior, execute synthesis to convert high-level descriptions to gate-level implementations, and manage physical layouts to ensure designs can be realized in silicon. As shown in Figure 1, the EDA design flow includes several key stages: logic synthesis, floorplanning, placement, Clock Tree Synthesis (CTS), and routing. Below are concise descriptions of each stage, illustrating their importance in the integrated circuit development process.

**Logic Synthesis** transforms a high-level circuit description into an optimized gate-level netlist (Berndt et al., 2022; Wang et al., 2024a;b). **Floorplan** involves deciding the layout of major components within an integrated circuit, positioning blocks and core components to balance signal integrity, power distribution, and area utilization. **Placement** involves assigning specific locations to various circuit components within the core area of the chip, following the floorplanning stage. The primary objective of this stage is to strategically place the components to optimize performance metrics such as delay and power consumption while ensuring adherence to design rules (Geng et al., 2024). **Clock Tree Synthesis (CTS)** creates a clock distribution network within an IC to minimize those clock effects, and ensure the correct timing synchronization for circuit operation. **Routing** involves creating the physical paths for electrical connectivity between various components on the IC as per the netlist. This stage must handle multiple layers of the chip, avoid obstacles, manage signal integrity, and meet all electrical and timing constraints (Cheng et al., 2022).

**Chip Placement** The placement stage is crucially divided into two distinct phases: *macro placement* and *cell placement*. (1) Macro placement is a critical very large-scale integration (VLSI) physical design problem that targets the arrangement of larger components, such as SRAMs and clock generators—often called macros. This phase significantly impacts the chip's overall floorplan and essential design parameters like wirelength, power, and area. (2) Following this, the standard cell placement phase addresses the arrangement of the more numerous and smaller standard cells, which serve as the fundamental building blocks of digital designs. This phase typically utilizes analytical solvers to secure an optimized configuration that not only minimizes wirelength but also enhances the electrical and timing performance of the chip.

## 4 DATASET

### 4.1 DESCRIPTION OF DESIGNS

Due to the oversimplification of datasets in early years, there exists a significant gap between these datasets and real-world applications. For instance, the usually used Bookshelf format (Nam et al., 2005; Adya et al., 2009) is overly simplified so that placement results given in such format are inapplicable for the subsequent stages to obtain a valid final design. Some later datasets (Kim et al., 2015) provide the LEF/DEF and necessary files for running these stages, but the contained circuits are still limited and they still lack some information for open-source tools like OpenROAD to work. For instance, the library file lacks buffer definitions, which is necessary for the clock tree synthesis phase, and the LEF file has incomplete layer definitions, which hinders the routing phase. To address this issue, we construct a dataset with comprehensive physical implementation information across the entire flow. Our dataset involves collecting a series of designs spanning various domains, including components such as CPUs, GPUs, network interfaces, image processing technologies, IoT

Table 2: Statistics of designs used in our benchmark.

| Id | Design | #Cells | #Nets | #Macros | #Pins | #IOs | #Edges |
|---|---|---|---|---|---|---|---|
| 1 | 8051 (lajanugen) | 13865 | 16424 | 0 | 50848 | 10 | 16174 |
| 2 | ariane136 (The-OpenROAD-Project) | 175248 | 191081 | 136 | 609834 | 495 | 187911 |
| 3 | ariane133 (The-OpenROAD-Project) | 168551 | 184856 | 132 | 592261 | 495 | 183142 |
| 4 | bp (The-OpenROAD-Project) | 301030 | 333364 | 24 | 984093 | 1198 | 333364 |
| 5 | bp_be (The-OpenROAD-Project) | 50881 | 58428 | 10 | 182949 | 3029 | 58092 |
| 6 | bp_fe (The-OpenROAD-Project) | 33206 | 36379 | 11 | 111510 | 2511 | 36203 |
| 7 | CAN-Bus (Tommydag) | 815 | 935 | 0 | 2637 | 13 | 935 |
| 8 | DE2_CCD_edge (suntodai) | 2333 | 3270 | 0 | 7823 | 64 | 3170 |
| 9 | dft48 Brendon Chetwynd, Kevin Bush, Kyle Ingols | 48488 | 52575 | 68 | 125501 | 132 | 50654 |
| 10 | FPGA-CAN (WangXuan95) | 140848 | 178913 | 0 | 532024 | 4 | 176472 |
| 11 | iot shield (brmcfarl) | 904 | 1006 | 0 | 2995 | 33 | 974 |
| 12 | mor1kx (openrisc, b) | 104293 | 130743 | 0 | 374983 | 576 | 125979 |
| 13 | or1200 openrisc (a) | 43386 | 32195 | 20 | 97047 | 383 | 31958 |
| 14 | OV7670_i2c (AngeloJacobo) | 332 | 340 | 0 | 979 | 29 | 316 |
| 15 | picorv (YosysHQ) | 8851 | 10531 | 0 | 32195 | 409 | 10470 |
| 16 | serv (olofk) | 1291 | 1482 | 0 | 3915 | 306 | 1403 |
| 17 | sha256 (secworks) | 10120 | 12283 | 0 | 38758 | 77 | 12176 |
| 18 | subrisc (Hara-Laboratory) | 859382 | 1103295 | 0 | 3359066 | 34 | 1092653 |
| 19 | swerv_wrapper (The-OpenROAD-Project) | 96435 | 105026 | 28 | 354652 | 1416 | 104565 |
| 20 | toygpu (matt-kimball) | 368081 | 466513 | 0 | 1399167 | 11 | 461675 |

devices, cryptographic units, and microcontrollers. Additionally, the dataset features a diverse array of sizes, with cell ranges from thousands to nearly a million. Our dataset features a complete design suite that supports the full EDA flow and includes a diverse range of sizes and domains, as illustrated in Table 1. The statistics for each case is detailed in Table 2, and we defer more details to Table 7 in Appendix C.2.

## 4.2 DATASET GENERATION PIPELINE

We use OpenROAD (Kahng & Spyrou, 2021), an open-source EDA tool, for generating our dataset. OpenROAD integrates various tools, such as Yosys (Wolf, 2016) for logic synthesis, TritonMacroPlacer for macro placement, RePlAce (Cheng et al., 2018) for cell placement, TritonCTS for clock tree synthesis, and TritonRoute for detailed routing. The choice of open-source tool allows for full reproducibility of our results and supports the promotion of the open-source community, ensuring that all generated data and methodologies are open-source. The initial dataset generation starts with Verilog files as raw data. OpenROAD performs logical synthesis to convert these high-level descriptions into a netlist, detailing the electrical connections among circuit components. This netlist is then used by OpenROAD's integrated floorplanning tool to configure the physical layout of the circuit on silicon. The resulting design from the floorplanning stage is converted into `LEF`/`DEF` files by OpenROAD, facilitating the application of subsequent placement algorithms. Simultaneously, we complete the EDA design flow through OpenROAD, generating data at subsequent stages, including placement, CTS, and routing.

## 5 ALGORITHMS

AI-based chip placement algorithms can be roughly grouped into three categories: black-box optimization (BBO) methods, analytical methods (gradient-based methods), and reinforcement learning (RL) methods. Each category frames the placement task as an optimization problem but adopts distinct objectives and methodologies. We present details as follows.

### 5.1 BLACK-BOX-OPTIMIZATION (BBO) METHODS

A straightforward intuition is to view the chip placement task as a black-box-optimization (BBO) problem, where the inner workings of the objective functions are inaccessible, and solutions are evaluated only based on the output metrics.

**Simulated Annealing (SA)** is a heuristic BBO optimization algorithm favored for its simplicity in implementation. Specifically, the SA algorithm generates solutions by perturbing the solution

space and then assessing the resulting representation. Different methods have been developed to effectively map representations to placement solutions (Kirkpatrick et al., 1983; Sherwani, 2012; Ho et al., 2004; Shunmugathammal et al., 2020; Vashisht et al., 2020), such as sequence pair (Murata et al., 1996) and B*-tree (Chang et al., 2000). Solutions are probabilistically accepted based on an annealing temperature to escape local optima in pursuit of a global optimum. Due to its simplicity in implementation, the SA algorithm often serves as a strong baseline in previous studies. In this work, we incorporate a specific SA implementation (Cheng et al., 2023) utilizing operations like swaps, shifts, and shuffles, and a cost function that balances wirelength, density, and congestion.

**WireMask-EA** (Shi et al., 2023) is a BBO framework that was recently introduced at the NeurIPS 2023 conference, positioning itself as an innovative approach in the intersection of AI and EDA. The framework utilizes a novel concept called wiremask, which plays a crucial role in guiding the mapping process from genotypes to phenotypes in a greedy manner. The wiremask concept was originally introduced by Lai et al. (2022), where it is defined as a matrix that predicts the potential increase in Half-Perimeter Wirelength (HPWL) for each subsequent macro placement on the design canvas. By estimating the wirelength increase, the wiremask helps in making informed decisions during the placement process, thereby potentially improving the quality of the layout. Building upon this concept, Shi et al. (2023) extended the framework to integrate several types of Black-Box Optimization (BBO) algorithms, including random search (RS), evolutionary algorithm (EA), and Bayesian optimization (BO), demonstrating the versatility of the approach in handling complex optimization tasks in chip design.

## 5.2 ANALYTICAL (GRADIENT-BASED) METHODS

Analytical methods formulate the optimization objective as an analytical function of module coordinates. This formulation enables efficient solutions through techniques like quadratic programming (Kahng et al., 2005; Viswanathan et al., 2007a;b; Spindler et al., 2008; Chen et al., 2008; Kim et al., 2012; Kim & Markov, 2012) and direct gradient descent (Lu et al., 2014; Cheng et al., 2018; Lin et al., 2019; 2020; Gu et al., 2020; Liao et al., 2022). This work focuses on the gradient-based algorithms, which are by far the more mainstream algorithms.

**DREAMPlace** (Liao et al., 2022) is a GPU-accelerated framework that leverages differentiable proxies, such as approximate HPWL, as optimization objectives. It was built upon the previous analytical placement algorithms, ePlace (Lu et al., 2014) and RePlAce (Cheng et al., 2018), yet significantly speeding up the placement process by using GPUs for acceleration. The series of versions of DREAMPlace introduces diverse differentiable proxies to better align the PPA improvement.

**AutoDMP** (Agnesina et al., 2023) extends DREAMPlace by automating hyperparameter tuning through multi-objective Bayesian optimization. It further accelerates the optimization process and reduces manual tuning efforts. At that time, this work showcased the promising potential of integrating GPU-accelerated algorithms with machine learning techniques for automating VLSI design.

## 5.3 REINFORCEMENT LEARNING (RL) METHODS

As VLSI systems grow in complexity, RL methods are being explored to enhance placement quality. GraphPlace (Mirhoseini et al., 2021) first models macro placement as a RL problem. Subsequently, DeepPR (Cheng & Yan, 2021) and PRNet (Cheng et al., 2022) establish a streamlined pipeline encompassing macro placement, cell placement, and routing. However, they treat density as a soft constraint, which may violate non-overlap constraint during training. Therefore, in this work, we mainly focus on MaskPlace and ChiPFormer, which are recent SOTA algorithms with hard non-overlapping constraints.

**MaskPlace** (Lai et al., 2022) represents the chip states as pixel-level visual inputs, including a wiremask (recording the HPWL increment for each grid), the viewmask (a global observation of the canvas), and the positionmask (to ensure non-overlapping constraint). Furthermore, it uses dense reward to boost the sample efficiency.

**ChiPFormer** (Lai et al., 2023) represents the first offline RL method. It is pretrained on various chips via offline RL and then fine-tuned on unseen chips for better efficiency. As a result, the time for placement is significantly reduced.

## 6 EVALUATION

### 6.1 EVALUATION METRICS

#### 6.1.1 FINAL DESIGN PPA METRICS

The primary goal of the entire Electronic Design Automation (EDA) workflow is to optimize the final PPA metrics. PPA stands for performance, power, and area—three crucial dimensions used to evaluate the quality of a chip product. These dimensions are assessed using several critical metrics, including worst negative slack (WNS), total negative slack (TNS), number of violating paths (NVP), power, and area. Optimizing these PPA metrics has been a major focus in the industry, approached through expert-designed heuristics. However, the challenge of PPA optimization has not been fully recognized within the AI community. Bridging this gap and improving the incorporation of AI strategies into PPA optimization are key goals of this benchmark.

In terms of specific metrics, Worst Negative Slack (WNS) and Total Negative Slack (TNS) are essential for assessing the timing performance of a chip circuit. Slack is the discrepancy between the expected and required arrival times of a signal, with negative slack indicating a timing violation. WNS pinpoints the most severe negative slack within a circuit, thus identifying the most critical timing issue. Conversely, TNS aggregates all negative slacks, providing a comprehensive view of the circuit's overall timing challenges. Moreover, the number of violating paths (NVP) counts the paths that fail to meet the timing constraints, further illustrating the timing performance issues.

#### 6.1.2 INTERMEDIATE METRICS

Commonly used intermediate surrogate metrics include Congestion, Wire Length (WL), Half Perimeter Wire Length (HPWL), and Macro HPWL (mHPWL). Congestion evaluates the density of wires in different chip regions. High congestion in certain areas can pose substantial challenges during the routing stage. While not a direct component of the PPA metrics, managing congestion effectively is essential to ensure that the chip can be successfully manufactured. Therefore, it is also considered as an evaluation metric in this paper. Congestion is typically estimated after the Clock Tree Synthesis (CTS) stages but before the detailed routing stage, allowing for adjusting macro placement and routing strategies to mitigate potential issues.

Wire Length (WL) is the total length of all wires connecting all modules in a chip. Half Perimeter Wire Length (HPWL) is the sum of half perimeters of bounding boxes that encompass all pins in each net. It is widely used as an estimation of WL and is obtained after cell placement. Macro HPWL (mHPWL) further simplifies HPWL by only considering the macros. It is favored in recent studies as it can be immediately obtained after macro placement. These metrics are thought to correlate with the final PPA, but they do not directly reflect the chip quality.

### 6.2 END-TO-END EVALUATION WORKFLOW

We present an end-to-end evaluation workflow utilizing OpenROAD-flow-scripts (Kahng & Spyrou, 2021) for the various stages of the EDA design flow, as illustrated in Figure 2. All tools used in this workflow are open-source, providing a significant advantage over other workflows that rely on commercial software. This workflow is designed to offer a comprehensive assessment of optimization algorithms at any stage of the design flow.

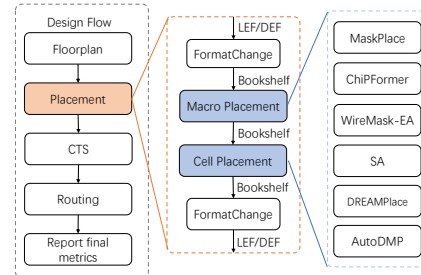

Our dataset comprises design kits needed for each stage of the physical design flow. For evaluating any stage-specific algorithm, the output file from the preceding stage serves as the input for the algorithm under evaluation. The algorithm processes this input to generate its output, which is subsequently plugged into the Open-ROAD design flow. Ultimately, final performance metrics

Figure 2: Illustration of our end-to-end evaluation workflow.

such as TNS, WNS, Area, and Power are reported, providing a comprehensive end-to-end performance assessment. This method offers a holistic set of metrics that can evaluate the optimization

Table 3: The evaluation results of AI-based macro placement algorithms. MacroHPWL, HPWL, Congestion are intermediate metrics, and the other metrics evaluate the end-to-end performance.

| Method | Intermediate Metrics | | | PPA Metrics | | | | | |
|--------|-----------|--------|-------------|-------------|--------|--------|--------|--------|--------|
| | MacroHPWL ↓ | HPWL ↓ | Congestion ↓ | Wirelength ↓ | Power ↓ | WNS ↓ | TNS ↓ | NVP ↓ | Area ↓ |
| WireMask-EA | **0.647** | 1.027 | 1.105 | 1.099 | 1.015 | 1.085 | 0.995 | 0.967 | 1.004 |
| SA | 0.836 | 1.044 | 1.099 | 1.097 | 1.062 | 1.121 | 1.311 | 1.109 | 1.013 |
| DREAMPlace | 0.857 | 0.974 | 1.049 | 1.059 | 1.015 | 1.112 | 1.025 | 1.038 | 0.999 |
| AutoDMP | 0.698 | **0.892** | **0.950** | **0.950** | 1.013 | 1.196 | 1.540 | 1.176 | 1.002 |
| MaskPlace | 1.681 | 1.119 | 1.148 | 1.148 | 1.051 | 1.014 | **0.978** | **0.903** | 1.014 |
| ChiPFormer | 0.681 | 0.976 | 1.027 | 1.024 | 1.015 | 1.031 | 1.355 | 1.223 | **0.981** |
| OpenROAD | 1.000 | 1.000 | 1.000 | 1.000 | **1.000** | **1.000** | 1.000 | 1.000 | 1.000 |

Table 4: The evaluation results of AI-based standard cell placement algorithms.

| Method | Intermediate Metrics | | PPA Metrics | | | | | |
|--------|--------|-------------|-------------|---------|--------|--------|--------|--------|
| | HPWL ↓ | Congestion ↓ | Wirelength ↓ | Power ↓ | WNS ↓ | TNS ↓ | NVP ↓ | Area ↓ |
| DREAMPlace | **0.981** | **0.999** | 1.008 | **0.987** | 1.321 | 4.678 | 5.313 | 0.996 |
| AutoDMP | 1.124 | 1.123 | 1.138 | 1.011 | 1.540 | 1.916 | 1.119 | **0.995** |
| OpenROAD | 1.000 | 1.000 | **1.000** | 1.000 | **1.000** | **1.000** | **1.000** | 1.000 |

effects of any stage-specific algorithm on the final chip design, providing consistency of metrics and avoiding the limitations of overly simplified metrics confined to a single stage. It is particularly beneficial for the optimization and development of various algorithms, ensuring their improvements translate to practical enhancements in chip design and foster the development of more efficient and effective open-source EDA tools through a robust framework for testing and improvement.

## 6.3 EXPERIMENTAL SETUP

We apply the aforementioned workflow to evaluate six macro placement algorithms: SA, WireMask-EA, DREAMPlace, AutoDMP, MaskPlace, ChiPFormer, and the default algorithm in OpenROAD. Additionally, we also assess cell placement algorithms for DREAMPlace and AutoDMP. As most of these methods only support the circuit data in a `BookShelf` format, while the circuits in our used dataset are in a standard `LEF/DEF`, we start by converting the `LEF/DEF` files from the floorplan stage of our dataset to `BookShelf` format to serve as the input for the placement algorithms. After finishing the macro placement stage, the resulting placement files are then converted back to `DEF` format and reintroduced into the original flow. The resulting placement files in `BookShelf` format are then converted back to `DEF` and reintroduced into the original flow. Finally, we report the final metrics, obtaining end-to-end evaluation results. Additionally, we perform global placement and detailed placement using OpenRoad's native Place method and complete the entire flow to obtain the final metrics for comparison with other algorithms. Our project is open-sourced on GitHub.

## 7 RESULTS AND DISCUSSIONS

### 7.1 MAIN RESULTS

**Macro Placement** We evaluate the AI-based chip placement algorithms, including SA, WireMask-EA, DREAMPlace, AutoDMP, MaskPlace, and ChiPFormer, using both intermediate metrics and end-to-end performance. The results for macro placement are in Table 3. ChiPFormer and WireMask-EA demonstrated a significant reduction in MacroHPWL compared to OpenROAD using TritonMacroPlacer. WireMask-EA achieved the best performance in terms of MacroHPWL. While these AI-based placement algorithms showed good performance on several intermediate metrics, they perform poorly in terms of the end-to-end metrics compared to OpenRoad, particularly in Power, TNS, and Area. This outcome revealed a significant gap between the originally optimized MacroHPWL intermediate metrics and the final design PPA.

**Cell Placement** As shown in Table 4, DREAMPlace achieved the best results in the intermediate metrics of HPWL, and performed well in terms of Power. However, OpenROAD achieved the

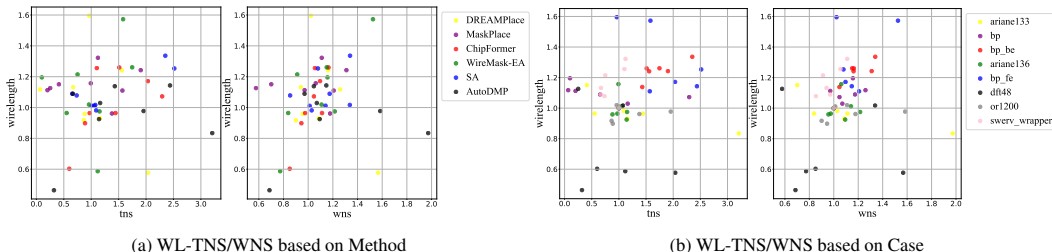

(a) WL-TNS/WNS based on Method          (b) WL-TNS/WNS based on Case

Figure 3: Correlations Between Wirelength and TNS/WNS, In the visualizations, points that share the same color represent data from (a) same method or (b) same case, respectively.

best results in WNS, TNS, and NVP, further demonstrating the inconsistency between intermediate metrics and final PPA results.

## 7.2  CORRELATION ANALYSIS

In this section, we conduct compute and discuss the correlation between the core optimization indicator MacroHPWL, used in existing placement algorithms, and the final chip performance metrics such as WNS, TNS, and wirelength, obtained through the OpenROAD process.

We use the Pearson correlation coefficient (Cohen et al., 2009) to evaluate the strength of linear correlation between pairs of metrics. The formula for calculating the Pearson correlation coefficient takes the form of

$$r = \frac{\sum (X_i - \overline{X})(Y_i - \overline{Y})}{\sqrt{\sum (X_i - \overline{X})^2 \sum (Y_i - \overline{Y})^2}}, \tag{1}$$

where $X_i$ and $Y_i$ are the observations, and $\overline{X}$ and $\overline{Y}$ are the respective means.

The results are shown in Figure 4. To calculate the correlation, the signs of all values are adjusted so that for all metrics the lower indicates the better. The results show that MacroHPWL only has a weak correlation with the Wirelength, which indicates that existing algorithms that optimize MacroHPWL do not lead to an optimization on the Wirelength. In contrast, HPWL shows a very strong positive correlation with actual Wirelength, indicating that HPWL works as an effective surrogate for approximating the Wirelength.

In addition to Wirelength, the final PPA metrics of the chip are associated with WNS/TNS, Area, and Power. The analyses shows that the correlation between MacroHPWL and these metrics is weak, indicating that optimization of MacroHPWL has minimal impact on these performance indicators. Moreover, the results in Figure 3 show that Wirelength exhibits weak correlations with WNS and TNS as well. This implies that even if a single-point algorithm successfully optimizes metrics such as Wirelength, the ultimate physical implementation might only enhance one aspect of the PPA metrics and may not effectively optimize the other dimensions. Therefore, more appropriate intermediate metrics are needed to better correlate with the actual PPA objectives.

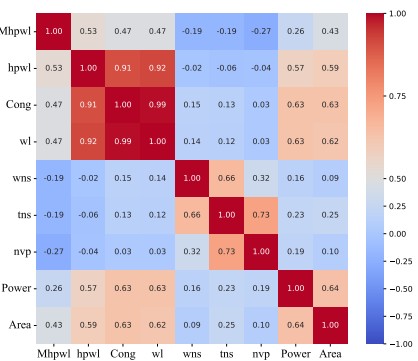

Figure 4: Correlation Between Metrics

## 7.3  FURTHER ANALYSIS

In this section, we analyze specific cases to assess the impact of different methods on PPA using ariane133 as the benchmark. Table 5 presents the experimental data, while Figure 5 shows the performance variations on the worst path caused by different placement algorithms. The AutoDMP method reduces wirelength but worsens timing, with smaller area and lower power. This is due to fewer buffers

Table 5: The evaluation results of ariane133 under AI-based macro placement algorithms.

| Method | Intermediate Metrics | | | PPA Metrics | | | | | |
|---|---|---|---|---|---|---|---|---|---|
| | MacroHPWL ↓ | HPWL ↓ | Congestion ↓ | Wirelength ↓ | Power ↓ | WNS ↑ | TNS ↑ | NVP ↓ | Area ↓ |
| WireMask-EA | 1124169 | 5065453.5 | 0.226 | 6583143 | 0.369 | -0.417 | -329.353 | 1970 | 349228 |
| SA | 1683330 | 5187015.8 | 0.230 | 6699484 | 0.365 | -0.512 | -650.399 | 2317 | 347043 |
| MaskPlace | 4444289 | 6253554.3 | 0.265 | 7853892 | 0.373 | **-0.349** | **-244.936** | **1636** | 357322 |
| ChiPFormer | 1253799 | 5019138.5 | 0.226 | 6581086 | 0.370 | -0.553 | -860.952 | 2703 | 349005 |
| DREAMPlace | 1111023 | 4826654.0 | 0.214 | 6348638 | 0.367 | -0.540 | -690.266 | 2307 | 348180 |
| AutoDMP | **828592** | **4250870.4** | **0.192** | **5694373** | **0.350** | -0.982 | -1913.660 | 3310 | **344091** |
| OpenROAD | 2685856 | 5260791.9 | 0.235 | 6825071 | 0.359 | -0.498 | -596.718 | 2274 | 352706 |

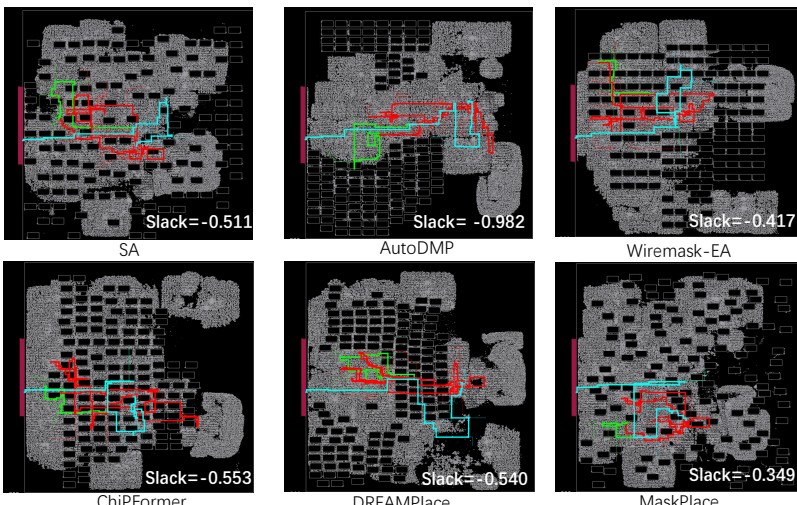

Figure 5: The image of the worst path for each method in ariane133.

being added during timing repair, leading to degraded timing despite the reduced area and power. Moreover, wirelength and timing are not always proportional—critical paths still experience delays, indicating that global wirelength optimization may miss timing-sensitive paths, especially in worst-case scenarios. For more details, see Appendix C.1.

### 7.4 DISCUSSION

Our benchmark comprises design kits essential for each stage of the physical design flow, including netlists, libraries, rules, and constraints needed during the physical implementation stage. This thorough inclusion allows for a convenient and detailed evaluation of algorithms at specific stages of the physical design flow, enabling researchers and practitioners to test and compare the effectiveness of their solutions in a realistic, end-to-end environment. We call on the AI researchers to pay more attention on the "shift-left" challenge from the real-world industrial scenarios, keeping towards the mission of bridging the huge gap between academic research and industrial applications.

## 8 CONCLUSION

This paper presents a comprehensive dataset that spans the entire spectrum of the EDA design process and an end-to-end evaluation method, which we used to assess several placement algorithms: SA, WireMask-EA, DREAMPlace, AutoDMP, MaskPlace, and ChiPFormer. Our evaluation revealed inconsistencies between the metrics currently emphasized by mainstream placement algorithms and the final performance outcomes. These findings highlight the need for a new perspective in the development of placement algorithms.

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

# A  TECHNICAL DETAILS

## A.1  EXPERIMENTAL DETAILS

In our workflow, it starts with OpenROAD. The file format for physical design in OpenROAD is OpenDB database (.odb), which contains `LEF/DEF` information. OpenROAD can be used to convert ODB to `LEF/DEF`, or `LEF/DEF` to ODB. Initially, the ODB after macro placement in the OpenROAD flow is converted to `LEF/DEF` files, serving as inputs for other macro placement algorithms. Since most existing macro placement algorithms only support bookshelf format input, DREAMPlace's code is used to convert `LEF/DEF` to bookshelf (.nodes, .pl, .nets) format. It is important to note that when DREAMPlace reads the DEF file, it treats the blockages in the BLOCKAGES part as virtual macros, so when using DREAMPlace's PlaceDB in Python to read macro information, the virtual macros must be excluded.

For ChiPFormer, use the default model and perform 100 online iterations to fine-tune the macro layout.For MaskPlace, run 3000 epochs.Other settings use the defaults.

In the method of ChipFormer and MaskPlace, the experiments were run on an NVIDIA GeForce RTX 2080 Ti, taking one day for all cases.For the other algorithms, we used 32 CPUs (Intel(R) Xeon(R) CPU E5-2667 v4 @ 3.20GHz), with a total time expenditure of two days.

Next, the macro layout (.pl) files are obtained after running ChiPFormer, MaskPlace, WireMask-EA, and SA, and then DREAMPlace is used to write the macro layout into DEF files. Since DREAMPlace does not modify the blockages in the DEF file, if blockages are defined in the DEF file, additional script modifications may be required.

After obtaining the macro-placed DEF files, they are converted to ODB using OpenROAD, followed by Tapcell and Welltie insertion, PDN generation, IO place, global place, detail place, and subsequent CTS and routing.

The performance of OpenROAD is limited; in the future, other global placement tools and detailed placement tools can be used to run the entire workflow.

## A.2  ENCOUNTERED ERRORS

Due to certain parsing bugs in OpenROAD, when converting DEF files to ODB files, it is necessary to ensure that no object names in the DEF file contain "/", otherwise, there will be issues during timing tests (issues will occur when writing and reading SPEF file).

Since various algorithms are academic and cannot directly set the minimum spacing between macros (which can be set in OpenROAD), the optimization of macro positions by these algorithms may result in macros being placed too closely. When integrating back into OpenROAD, the following errors might occur:

OpenROAD's limited capabilities in various stages also contribute to these errors. During PDN, "Unable to repair all channels." might be encountered. During the global place stage, global placement might diverge. During the detail place stage, detailed placement might fail (because the surrounding space of certain cells makes it impossible for OpenROAD to find space for adjustments).

The code of ChiPFormer has certain issues and needs modifications to be applied to other cases. DREAMPlace directly uses the mixed-size method with LEF/DEF for running macros.

For the ICCAD 2015 dataset, it includes the lib file, LEF/DEF files, netlist file, and sdc file. However, the lib file lacks buffer definitions, preventing CTS. The lef files have incomplete layer definitions, hindering routing.Additionaly, OpenRoad does not support the syntax of the ICCAD 2015 sdc file.

# B  LIMITATION

Our dataset currently has limitations in terms of data volume. In the future, we aim to increase the dataset's size and include more cases from various domains. This will enhance its generalizability and robustness, making it a more comprehensive and valuable resource for researchers . By ex-

Table 6: Comparison of the worst path wirelength, delay, and slack for various methods on the ariane133 layout.

| Method | WireMask-EA | SA | MaskPlace | ChiPFormer | DREAMPlace | AutoDMP | OpenROAD |
|---|---|---|---|---|---|---|---|
| wirelength | 3892 | 4107 | 3529 | **5761** | 4612 | 4318 | 4121 |
| delay | 4.36 | 4.57 | 4.25 | 4.49 | 4.54 | **4.92** | 4.44 |
| slack | -0.42 | -0.54 | -0.35 | -0.55 | -0.54 | **-0.98** | -0.51 |

panding the dataset, we will improve its applicability across different areas of chip design and EDA, further refining and perfecting this important resource.

## C   MORE RESULTS

### C.1   ADDITIONAL ANALYSIS

In this section, we provide a detailed analysis of the layout results for the ariane133 benchmark, focusing on the observed relationship between wirelength, timing, area, and power. The AutoDMP method shows interesting yet seemingly contradictory outcomes—reduced wirelength with worse timing, smaller area, and lower power. To understand these results, we explore the factors that contribute to these trends and propose potential directions for placement algorithm improvements.

A key factor influencing timing in large-scale circuits is buffer insertion. In general, when the wirelength between two pins is large, signal transmission delay increases. Buffer insertion helps by breaking long interconnects into smaller segments, reducing the overall transmission delay. Furthermore, buffers enhance driving capacity by reducing the load on the signal driver, which helps lower signal delay and improve timing. However, fewer buffers result in smaller area and lower power but also lead to degraded timing performance. In the case of ariane133, the AutoDMP method inserts fewer buffers during timing repair, resulting in smaller overall area and power consumption. However, the lack of sufficient buffers leads to worse timing outcomes, as seen in the experimental data. This explains why, despite achieving minimized wirelength, the timing performance does not improve accordingly.

The relationship between wirelength and timing is often assumed to be directly proportional; however, the analysis of ariane133 demonstrates that this is not always the case. Although the total wirelength (as indicated by the HPWL) is reduced, the timing performance does not improve proportionally. In fact, congestion is lower and wirelength is minimized, but critical paths still suffer from significant delays. This suggests that global wirelength optimization does not always lead to improved timing, particularly for worst-case paths. Table 6 provides insights into this phenomenon by highlighting the inconsistencies between wirelength and timing slack on the worst paths. Specifically, the data reveals that longer wirelength paths do not always exhibit worse timing, indicating that the optimization strategy may not have sufficiently accounted for timing-sensitive paths.

To address these issues and improve overall placement quality, several optimization strategies can be explored. One potential solution to mitigate timing degradation is to apply weighted optimization that emphasizes critical paths. By giving more weight to timing-sensitive paths during wirelength reduction, it would be possible to achieve better alignment between wirelength optimization and timing improvement. Another approach is to adopt mixed-size placement techniques that optimize both macros and standard cells simultaneously. Since macro wirelength (macro HPWL) has a weaker correlation with backend performance metrics, this approach could lead to a more balanced placement outcome, particularly for designs where macro placement significantly impacts PPA. Additionally, machine learning models that predict PPA based on placement features could provide valuable insights during placement optimization. These models would allow the algorithm to dynamically adjust its strategy based on predicted PPA outcomes, leading to more informed trade-offs between PPA.

### C.2   RAW DATA

All the raw data from the experiment are in Tables 8-27.

Table 7: Detailed descriptions of our collected designs.

| Id | Design | Description |
|----|--------|-------------|
| 1 | 8051lajanugen | FPGA implementation of the 8051 Microcontroller |
| 2 | ariane133 The-OpenROAD-Project | Ariane Core |
| 3 | ariane136 The-OpenROAD-Project | Ariane Core |
| 4 | bp The-OpenROAD-Project | Full 64-bit RISC-V Core with Cache Coherence Directory |
| 5 | bp_be The-OpenROAD-Project | Back-end of a 64-bit RISC-V Core with Cache Coherence Directory |
| 6 | bp_fe The-OpenROAD-Project | Front-end of a 64-bit RISC-V Core with Cache Coherence Directory |
| 7 | CAN-BusTommydag | A CAN bus Controller |
| 8 | DE2_CCD_edge suntodai | Image processing |
| 9 | dft48Brendon Chetwynd, Kevin Bush, Kyle Ingols | DFT design |
| 10 | FPGA-CANWangXuan95 | A lightweight CAN bus controller |
| 11 | iot shieldbrmcfarl | IoT Shield for the Intel Galileo Development Board |
| 12 | mor1kx openrisc (b) | an OpenRISC processor IP core |
| 13 | or1200openrisc (a) | OpenRISC 1200 implementation |
| 14 | OV7670_i2c AngeloJacobo | Camera interface |
| 15 | picorv YosysHQ | A Size-Optimized RISC-V CPU |
| 16 | serv olofk | An award-winning bit-serial RISC-V core |
| 17 | sha256secworks | Hardware implementation of the SHA-256 cryptographic hash function |
| 18 | subriscHara-Laboratory | Simple Instruction-Set Computer for IoT edge devices |
| 19 | swerv_wrapper The-OpenROAD-Project | SweRV RISC-V Core 1.1 from Western Digital |
| 20 | toygpu matt-kimball | A simple GPU on a TinyFPGA BX |

Table 8: The results of ariane133

| Method | Intermediate Metrics | | | PPA Metrics | | | | | |
|--------|--------------|--------|--------------|--------------|---------|---------|-----------|--------|----------|
| | MacroHPWL ↓ | HPWL ↓ | Congestion ↓ | Wirelength ↓ | Power ↓ | WNS ↑ | TNS ↑ | NVP ↓ | Area ↓ |
| WireMask-EA | 1124169 | 5065453.5 | 0.226 | 6583143 | 0.369 | -0.417 | -329.353 | 1970 | 349228 |
| SA | 1683330 | 5187015.8 | 0.230 | 6699484 | 0.365 | -0.512 | -650.399 | 2317 | 347043 |
| MaskPlace | 4444289 | 6253554.3 | 0.265 | 7853892 | 0.373 | **-0.349** | **-244.936** | **1636** | 357322 |
| ChiPFormer | 1253799 | 5019138.5 | 0.226 | 6581086 | 0.370 | -0.553 | -860.952 | 2703 | 349005 |
| DREAMPlace | 1111023 | 4826654.0 | 0.214 | 6348638 | 0.367 | -0.540 | -690.266 | 2307 | 348180 |
| AutoDMP | **828592** | **4250870.4** | **0.192** | **5694373** | **0.350** | -0.982 | -1913.660 | 3310 | **344091** |
| OpenROAD | 2685856 | 5260791.9 | 0.235 | 6825071 | 0.359 | -0.498 | -596.718 | 2274 | 352706 |

# D    LICENSE

The code and propose dataset will be publicly accessible. We include the following licenses for the raw data we used in this paper.

- CAN-Bus : MIT
- FPGA-CAN :GPL-3.0
- sha256:BSD-2-Clause
- DE2_CCD_edge:MIT
- picorv:ISC
- serv:ISC
- mor1kx:CERN-OHL-W
- ariane133:SOLDERPAD HARDWARE
- ariane136:SOLDERPAD HARDWARE
- bp:BSD-3-Clause
- bp_be:BSD-3-Clause
- bp_fe:BSD-3-Clause
- swerv_wrapper:Apache

Table 9: The results of ariane136

| Method | Intermediate Metrics | | | PPA Metrics | | | | | |
|---|---|---|---|---|---|---|---|---|---|
| | MacroHPWL ↓ | HPWL ↓ | Congestion ↓ | Wirelength ↓ | Power ↓ | WNS ↑ | TNS ↑ | NVP ↓ | Area ↓ |
| WireMask-EA | 1181404 | 5278467.4 | 0.239 | 6945252 | 0.570 | -1.726 | -4268.610 | 3628 | 358740 |
| SA | 1693986 | 5608216.9 | 0.248 | 7208805 | 0.571 | -1.620 | -3917.960 | 3487 | 359532 |
| MaskPlace | 4586233 | 6544926.4 | 0.285 | 8245218 | 0.583 | -1.508 | -3677.480 | 3966 | 363704 |
| ChiPFormer | 1269027 | 5244085.2 | 0.237 | 6869186 | 0.566 | -1.385 | -3603.530 | 3609 | 358773 |
| DREAMPlace | 1067974 | 5202240.5 | 0.231 | 6831531 | 0.571 | **-1.358** | **-3269.220** | 4350 | **356706** |
| AutoDMP | **870515** | **4993936.2** | **0.228** | **6590972** | 0.562 | -1.553 | -4236.020 | 4431 | 356714 |
| OpenROAD | 3067334 | 5561870.0 | 0.246 | 7124942 | **0.546** | -1.420 | -3707.930 | **3473** | 360120 |

Table 10: The results of bp

| Method | Intermediate Metrics | | | PPA Metrics | | | | | |
|---|---|---|---|---|---|---|---|---|---|
| | MacroHPWL ↓ | HPWL ↓ | Congestion ↓ | Wirelength ↓ | Power ↓ | WNS ↑ | TNS ↑ | NVP ↓ | Area ↓ |
| WireMask-EA | **112560** | 7526021.6 | 0.437 | 10002159 | 0.254 | -1.936 | -21.812 | 326 | 490622 |
| SA | 151696 | 6927899.3 | 0.398 | 9113560 | 0.253 | -1.965 | -143.317 | 969 | 489850 |
| MaskPlace | 324368 | 7135790.7 | 0.407 | 9307133 | 0.253 | **-1.625** | -44.217 | 478 | 489867 |
| ChiPFormer | 138943 | 6857073.6 | 0.392 | 8970666 | 0.250 | -1.752 | -502.977 | 2179 | 490305 |
| DREAMPlace | 161581 | 6989990.7 | 0.409 | 9347541 | 0.250 | -2.108 | **-14.609** | **192** | 489693 |
| AutoDMP | 135090 | 6568910.6 | 0.376 | 8610145 | **0.247** | -1.791 | -254.786 | 1821 | 486445 |
| OpenROAD | 119123 | **6389902.6** | **0.365** | **8365647** | 0.247 | -1.674 | -219.229 | 1095 | **484494** |

Table 11: The results of bp_be

| Method | Intermediate Metrics | | | PPA Metrics | | | | | |
|---|---|---|---|---|---|---|---|---|---|
| | MacroHPWL ↓ | HPWL ↓ | Congestion ↓ | Wirelength ↓ | Power ↓ | WNS ↑ | TNS ↑ | NVP ↓ | Area ↓ |
| WireMask-EA | 66729 | 2678100.1 | 0.524 | 3574875 | 0.467 | -2.142 | -4093.970 | 5131 | 123966 |
| SA | 85984 | 2568796.6 | 0.557 | 3788559 | 0.459 | -2.494 | -5510.140 | 5381 | 124667 |
| MaskPlace | 114314 | 2548230.4 | 0.513 | 3522611 | 0.456 | -2.439 | -4459.870 | 5173 | 124938 |
| ChiPFormer | **62168** | 2525703.6 | 0.524 | 3572070 | 0.425 | -2.169 | -3541.820 | 5100 | **110904** |
| DREAMPlace | 87304 | 2533359.4 | 0.516 | 3518916 | 0.458 | -2.163 | -3648.020 | 6026 | 125221 |
| AutoDMP | 86263 | 2539005.8 | 0.468 | 3227167 | 0.453 | -1.948 | -3351.120 | 5045 | 125471 |
| OpenROAD | 76561 | **2370091.6** | **0.409** | **2835542** | **0.409** | **-1.862** | **-2343.290** | **5013** | 118786 |

Table 12: The results of bp_fe

| Method | Intermediate Metrics | | | PPA Metrics | | | | | |
|---|---|---|---|---|---|---|---|---|---|
| | MacroHPWL ↓ | HPWL ↓ | Congestion ↓ | Wirelength ↓ | Power ↓ | WNS ↑ | TNS ↑ | NVP ↓ | Area ↓ |
| WireMask-EA | 45985 | 1863651.1 | 0.510 | 2783740 | 0.309 | -1.666 | -777.420 | 2628 | 77648.300 |
| SA | 59872 | 1603778.9 | 0.404 | 2218668 | 0.310 | -1.179 | -1236.880 | 2990 | 81919.500 |
| MaskPlace | 64024 | 1540944.3 | 0.355 | 1965209 | 0.309 | -1.318 | -771.363 | 2582 | 77881.100 |
| ChiPFormer | **45754** | 1544281.0 | 0.375 | 2073376 | 0.299 | -1.197 | -1000.190 | 2714 | **72723.900** |
| DREAMPlace | 60279 | 1975681.6 | 0.508 | 2823861 | 0.294 | -1.117 | **-473.261** | **1849** | 77427.800 |
| AutoDMP | 56107 | 1560612.6 | 0.365 | 2023847 | 0.312 | -1.253 | -1198.430 | 2626 | 81294.400 |
| OpenROAD | 53842 | **1468799.8** | **0.319** | **1770176** | **0.285** | **-1.092** | -491.698 | 1870 | 75884.500 |

Table 13: The results of dft48

| Method | Intermediate Metrics | | | PPA Metrics | | | | | |
|---|---|---|---|---|---|---|---|---|---|
| | MacroHPWL ↓ | HPWL ↓ | Congestion ↓ | Wirelength ↓ | Power ↓ | WNS ↑ | TNS ↑ | NVP ↓ | Area ↓ |
| WireMask-EA | 335246296 | 1039236.4 | 0.114 | 1330932 | 0.239 | -0.510 | -439.378 | 2035 | 81353.700 |
| SA | 339003863 | 1815564.2 | 0.198 | 2305190 | 0.304 | -0.882 | -422.568 | 1546 | 84170.100 |
| MaskPlace | 1409226442 | 2117408.1 | 0.223 | 2552954 | 0.258 | **-0.381** | **-98.916** | **548** | 83185.400 |
| ChiPFormer | 431188529 | 1094111.0 | 0.119 | 1366989 | 0.242 | -0.561 | -235.601 | 974 | 80770.400 |
| DREAMPlace | 267700115 | 800165.5 | 0.112 | 1308094 | 0.234 | -1.034 | -801.257 | 2274 | **79407.400** |
| AutoDMP | 270017585 | **774418.4** | **0.091** | **1051008** | **0.232** | -0.453 | -125.948 | 797 | 80201.100 |
| OpenROAD | 752297935 | 1981412.2 | 0.198 | 2267002 | 0.257 | -0.660 | -393.111 | 1411 | 84300.200 |

Table 14: The results of or1200

| Method | Intermediate Metrics | | | PPA Metrics | | | | | |
|---|---|---|---|---|---|---|---|---|---|
| | MacroHPWL ↓ | HPWL ↓ | Congestion ↓ | Wirelength ↓ | Power ↓ | WNS ↑ | TNS ↑ | NVP ↓ | Area ↓ |
| WireMask-EA | 102261063 | 1196931.6 | 0.035 | 1432301 | **0.049** | -1.417 | -1530.180 | **1754** | 66547.600 |
| SA | 113496323 | 1200696.8 | 0.034 | 1420640 | 0.059 | -1.294 | -1613.410 | 2674 | 66937.300 |
| MaskPlace | 208736767 | 1166784.1 | 0.033 | 1350820 | 0.059 | -1.505 | -2198.370 | 2684 | 66953.500 |
| ChiPFormer | **95035903** | 1089714.1 | **0.030** | **1262132** | 0.057 | -1.208 | -1415.960 | 2669 | 66256.900 |
| DREAMPlace | 153596940 | 1099937.2 | 0.031 | 1288746 | 0.057 | **-1.151** | **-1380.980** | 2670 | 65981.000 |
| AutoDMP | 122473700 | **1058807.4** | 0.033 | 1373410 | 0.058 | -2.025 | -3119.280 | 2760 | **65756.800** |
| OpenROAD | 187077195 | 1223792.9 | 0.034 | 1405193 | 0.060 | -1.278 | -1595.730 | 2677 | 67276.500 |

Table 15: The results of swerv_wrapper

| Method | Intermediate Metrics | | | PPA Metrics | | | | | |
|---|---|---|---|---|---|---|---|---|---|
| | MacroHPWL ↓ | HPWL ↓ | Congestion ↓ | Wirelength ↓ | Power ↓ | WNS ↑ | TNS ↑ | NVP ↓ | Area ↓ |
| WireMask-EA | 474447 | 3893655.8 | 0.414 | 4854661 | 0.666 | -1.027 | -873.506 | 1518 | 205627 |
| SA | 630919 | 3811960.2 | 0.368 | 4310596 | 0.648 | **-0.962** | -854.737 | 1811 | **200280** |
| MaskPlace | 1270726 | 4193788.0 | 0.452 | 5286392 | 0.690 | -1.251 | -1306.460 | 1696 | 206481 |
| ChiPFormer | **437638** | 4060188.1 | 0.428 | 5019849 | 0.674 | -1.189 | -1282.240 | 2139 | 205775 |
| DREAMPlace | 865544 | 3575327.7 | 0.366 | 4525348 | 0.646 | -1.061 | -780.196 | 1608 | 203896 |
| AutoDMP | 450662 | 3213155.0 | 0.372 | 4354901 | 0.648 | -1.094 | **-773.339** | 1529 | 200823 |
| OpenROAD | 666704 | **3177043.0** | **0.339** | **3997163** | **0.628** | -1.127 | -1163.820 | **1515** | 202853 |

Table 16: The results of 8051-master

| Method | Intermediate Metrics | | PPA Metrics | | | | | |
|---|---|---|---|---|---|---|---|---|
| | HPWL ↓ | Congestion ↓ | Wirelength ↓ | Power ↓ | WNS ↑ | TNS ↑ | NVP ↓ | Area ↓ |
| DREAMPlace | 142162.6 | 0.201 | 210127 | 0.075 | -0.634 | -16.460 | **35** | 29484.200 |
| AutoDMP | **131174.7** | **0.186** | **195536** | 0.074 | -0.633 | **-15.643** | **35** | **29365.600** |
| OpenROAD | 141245.3 | 0.201 | 207737 | **0.073** | **-0.603** | -16.664 | 41 | 29461.400 |

Table 17: The results of CAN-Bus

| Method | Intermediate Metrics | | PPA Metrics | | | | | |
|---|---|---|---|---|---|---|---|---|
| | HPWL ↓ | Congestion ↓ | Wirelength ↓ | Power ↓ | WNS ↑ | TNS ↑ | NVP ↓ | Area ↓ |
| DREAMPlace | **3916.8** | 0.093 | 5683 | **0.005** | -0.074 | -0.074 | **1** | 1675.270 |
| AutoDMP | 4210.8 | 0.097 | 6024 | 0.005 | **-0.074** | **-0.074** | **1** | 1668.880 |
| OpenROAD | 3955.1 | **0.092** | **5623** | 0.005 | -0.079 | -0.079 | **1** | **1665.430** |

Table 18: The results of DE2_CCD_edge

| Method | Intermediate Metrics | | PPA Metrics | | | | | |
|---|---|---|---|---|---|---|---|---|
| | HPWL ↓ | Congestion ↓ | Wirelength ↓ | Power ↓ | WNS ↑ | TNS ↑ | NVP ↓ | Area ↓ |
| DREAMPlace | **16556.5** | 0.123 | 23453 | 0.201 | -0.924 | -184.592 | 736 | 5797.470 |
| AutoDMP | 22203.0 | 0.160 | 30745 | 0.211 | -0.930 | -200.125 | 710 | 5812.630 |
| OpenROAD | 16689.0 | **0.120** | **22746** | **0.199** | **-0.919** | **-165.009** | **609** | **5758.100** |

Table 19: The results of FPGA-CAN

| Method | Intermediate Metrics | | PPA Metrics | | | | | |
|---|---|---|---|---|---|---|---|---|
| | HPWL ↓ | Congestion ↓ | Wirelength ↓ | Power ↓ | WNS ↑ | TNS ↑ | NVP ↓ | Area ↓ |
| DREAMPlace | 1472841.9 | 0.181 | 2437446 | **0.738** | -1.027 | -678.865 | 17688 | **378643** |
| AutoDMP | **1437916.7** | **0.178** | **2393640** | 0.745 | -0.892 | -108.201 | 3864 | 379675 |
| OpenROAD | 1483498.8 | 0.180 | 2394544 | 0.742 | **-0.404** | **-30.331** | **1843** | 379193 |

Table 20: The results of OV7670_i2c

| Method | Intermediate Metrics | | PPA Metrics | | | | | |
|---|---|---|---|---|---|---|---|---|
| | HPWL ↓ | Congestion ↓ | Wirelength ↓ | Power ↓ | WNS ↑ | TNS ↑ | NVP ↓ | Area ↓ |
| DREAMPlace | 733113.0 | 0.215 | 1164763 | 0.625 | -0.579 | -82.237 | 3154 | 157496 |
| AutoDMP | **701777.0** | **0.201** | **1091553** | **0.612** | -0.561 | **-23.030** | **1053** | **155624** |
| OpenROAD | 723012.1 | 0.208 | 1127120 | 0.617 | **-0.466** | -30.612 | 2327 | 156210 |

Table 21: The results of iot_shield

| Method | Intermediate Metrics | | PPA Metrics | | | | | |
|---|---|---|---|---|---|---|---|---|
| | HPWL ↓ | Congestion ↓ | Wirelength ↓ | Power ↓ | WNS ↑ | TNS ↑ | NVP ↓ | Area ↓ |
| DREAMPlace | 5014.3 | **0.118** | **7346** | **0.006** | -0.167 | -2.457 | 27 | **1712.770** |
| AutoDMP | 5976.2 | 0.137 | 8653 | 0.006 | **-0.152** | -2.367 | 29 | 1730.600 |
| OpenROAD | 5067.3 | 0.119 | 7349 | 0.006 | -0.152 | **-2.186** | **20** | 1719.420 |

Table 22: The results of mor1kx

| Method | Intermediate Metrics | | PPA Metrics | | | | | |
|--------|-----------|-------------|-------------|---------|--------|--------------|--------|---------|
| | HPWL ↓ | Congestion ↓ | Wirelength ↓ | Power ↓ | WNS ↑ | TNS ↑ | NVP ↓ | Area ↓ |
| **DREAMPlace** | **1723145.8** | **0.366** | **3421791** | **0.703** | -1.863 | -19564.300 | 26568 | 289570 |
| **AutoDMP** | 2258941.0 | 0.524 | 4893199 | 0.735 | -3.879 | -44135.900 | 26600 | **285455** |
| **OpenROAD** | 2006747.0 | 0.393 | 3661548 | 0.703 | **-0.796** | **-4928.920** | **25509** | 313065 |

Table 23: The results of picorv32

| Method | Intermediate Metrics | | PPA Metrics | | | | | |
|--------|-----------|-------------|-------------|---------|--------|--------------|--------|---------|
| | HPWL ↓ | Congestion ↓ | Wirelength ↓ | Power ↓ | WNS ↑ | TNS ↑ | NVP ↓ | Area ↓ |
| **DREAMPlace** | 88407.3 | **0.194** | 135544 | **0.048** | -0.147 | **-0.521** | **29** | **18951.400** |
| **AutoDMP** | 132458.3 | 0.244 | 170435 | 0.050 | -0.142 | -0.525 | 37 | 19039.000 |
| **OpenROAD** | **87168.8** | 0.195 | **134211** | 0.048 | **-0.107** | -0.661 | 38 | 18972.700 |

Table 24: The results of serv

| Method | Intermediate Metrics | | PPA Metrics | | | | | |
|--------|-----------|-------------|-------------|---------|--------|--------------|--------|---------|
| | HPWL ↓ | Congestion ↓ | Wirelength ↓ | Power ↓ | WNS ↑ | TNS ↑ | NVP ↓ | Area ↓ |
| **DREAMPlace** | 7194.0 | 0.165 | 11128 | 0.007 | **-0.563** | -10.333 | 81 | 2125.870 |
| **AutoDMP** | 10361.9 | 0.261 | 17455 | 0.008 | -0.593 | -11.893 | 120 | 2193.170 |
| **OpenROAD** | **7165.0** | **0.163** | **10638** | **0.007** | -0.565 | **-10.210** | **73** | **2097.940** |

Table 25: The results of sha256

| Method | Intermediate Metrics | | PPA Metrics | | | | | |
|--------|-----------|-------------|-------------|---------|--------|--------------|--------|---------|
| | HPWL ↓ | Congestion ↓ | Wirelength ↓ | Power ↓ | WNS ↑ | TNS ↑ | NVP ↓ | Area ↓ |
| **DREAMPlace** | 114376.4 | 0.203 | 170230 | **0.161** | -0.542 | -9.367 | 40 | 23476.400 |
| **AutoDMP** | 120832.7 | 0.206 | 173822 | 0.172 | -0.573 | -9.891 | **39** | **23398.700** |
| **OpenROAD** | **113841.8** | **0.202** | **168199** | 0.162 | **-0.528** | **-8.667** | **39** | 23426.900 |

Table 26: The results of subrisc

| Method | Intermediate Metrics | | PPA Metrics | | | | | |
|--------|-----------|-------------|-------------|---------|--------|--------------|--------|---------|
| | HPWL ↓ | Congestion ↓ | Wirelength ↓ | Power ↓ | WNS ↑ | TNS ↑ | NVP ↓ | Area ↓ |
| **DREAMPlace** | 13920769.0 | 0.264 | 21051063 | 2.774 | -0.847 | -5742.680 | 29283 | 2341490 |
| **AutoDMP** | **13446106.8** | **0.238** | **19114559** | **2.697** | **-0.673** | **-150.178** | **466** | **2319150** |
| **OpenROAD** | 14386152.9 | 0.253 | 20108941 | 3.425 | -0.782 | -305.819 | 674 | 2334330 |

Table 27: The results of toygpu

| Method | Intermediate Metrics | | PPA Metrics | | | | | |
|--------|-----------|-------------|-------------|---------|--------|--------------|--------|---------|
| | HPWL ↓ | Congestion ↓ | Wirelength ↓ | Power ↓ | WNS ↑ | TNS ↑ | NVP ↓ | Area ↓ |
| **DREAMPlace** | **4347733.8** | 0.198 | 6784639 | **1.101** | -1.503 | -131.661 | **101** | 979293 |
| **AutoDMP** | 4444860.0 | **0.194** | **6595166** | 1.109 | -2.418 | -201.797 | 107 | **967780** |
| **OpenROAD** | 4657703.6 | 0.212 | 7185236 | 1.108 | **-1.294** | **-103.668** | **101** | 981911 |

