# OpenReview forum: "Benchmarking End-To-End Performance of AI-Based Chip Placement Algorithms"
_ICLR.cc/2025/Conference — ICLR 2025 Conference Withdrawn Submission_

### Official Review · Reviewer_2tkx · 2024-10-22

**Soundness:** 2
**Presentation:** 4
**Contribution:** 1
**Rating:** 3
**Confidence:** 5

**Summary:**

This article describes a process that starts with Verilog files and goes through logic synthesis, floorplanning, placement, CTS, and routing stages to ultimately generate routed metrics, which are used as a dataset. The authors use the open-source ASIC chip design framework OpenROAD and replace the algorithm of the placement stage in their experiments to evaluate the differences in intermediate surrogate metrics and final PPA results among different placement algorithms.

**Strengths:**

- The writing is clear and helps in understanding the motivation and solution.
- The process of constructing the benchmark is detailed.
- The orientation is clear. It is important for the AI community to obtain PPA and other metrics in chip design tasks.

**Weaknesses:**

- The contribution of the paper is moderate. In the chip design field, the main issue for the AI community is not the inability to obtain PPA metrics, but the need to obtain them **quickly**. In fact, the placement results on datasets like CircuitNet 2.0 can be fully evaluated using either open-source tools like OpenROAD or commercial tools. This workflow is already streamlined (in essence, this part of the contribution comes from OpenROAD, not from this work itself). This work has facilitated the format exchange process between LEF/DEF and Bookshelf, but its contribution is limited.
- The conclusion of the paper is trivial. The mismatch between intermediate surrogate metrics and subsequent stages' intermediate surrogate metrics is a trivial phenomenon. In the EDA flow, intermediate surrogate metrics are used to **quickly** estimate the indicators of later stages in earlier stages. For example, HPWL used in the placement stage is a first-order approximate estimate for subsequent delay. Generally speaking, the closer the metric is to the subsequent stage, the more accurate it is; however, the evaluation cost is also higher.
- The choice of data is unremarkable. The authors need to explain why researchers from the AI or EDA community should choose the benchmark proposed by the authors. For example, the superblue series from ICCAD2015 is characterized by its large scale, the ISPD2015 dataset features fence regions, and CircuitNet 2.0 provides multimodal data that can help AI researchers reduce the time running commercial tools.

**Questions:**

- Will the benchmark data be open-sourced?
- Can the authors provide the platform configuration (such as GPU, CPU, etc.) in the experiment section, as well as the total runtime of the evaluation scripts and the runtime of each placer?

---

### Official Review · Reviewer_FVwT · 2024-10-30

**Soundness:** 2
**Presentation:** 2
**Contribution:** 1
**Rating:** 1
**Confidence:** 1

**Summary:**

The paper presents ChipBench, a benchmark specifically designed to evaluate the effectiveness
of existing AI-based chip placement algorithms.

**Strengths:**

The paper presents a new benchmark suite to evaluate existing AI-based chip placement algorithms, which is probably essential for the chip-design community.

**Weaknesses:**

- The paper isn’t relevant to the ICLR community, it is mostly relevant to the chip-design community.
- The paper violates ICLR’s double-blind requirement. The project is open-sourced in GitHub (https://github.com/MIRALab-USTC/ChiPBench), and the repo isn’t anonymous.

**Questions:**

None

---

### Official Review · Reviewer_tHx4 · 2024-10-31

**Soundness:** 3
**Presentation:** 3
**Contribution:** 2
**Rating:** 5
**Confidence:** 3

**Summary:**

This paper introduces a benchmark suite designed to evaluate AI-based chip placement algorithms, incorporating 20 circuit designs compiled from Verilog. The authors evaluate six algorithms—covering black-box optimization (BBO), gradient-based, and reinforcement learning (RL) approaches—against this benchmark for end-to-end performance. A key finding from the evaluation is the weak correlation between intermediate surrogate metrics and final design metrics (such as power, performance, and area, or PPA), highlighting the need for more reliable metrics. The benchmark suite, made openly available, represents a valuable resource for both academic and industry researchers, featuring diverse design characteristics across application domains, sizes, and configurations.

**Strengths:**

* The benchmark provides a well-rounded set of diverse circuit designs, covering various application domains, sizes, and characteristics, making it highly useful for generalizable evaluation of AI-driven placement algorithms.
* The open-source nature encourage wider adoption and enabling researchers to build on this foundation for further improvements in AI-based chip design.
* The evaluation spans multiple AI-based algorithmic approaches, making the suite a valuable comparative tool.
* By revealing the limited correlation between intermediate metrics and final PPA results, that may guide future research toward more effective and reliable evaluation criteria.

**Weaknesses:**

* The paper could benefit from more detailed analysis comparing each algorithm’s strengths and weaknesses on various design types.
* The benchmark’s 20-circuit scale, while diverse, may not fully capture the vast complexity of real-world designs. Discussion of potential extensions or adaptability to larger, more complex circuits would be useful for extending its practical impact.

**Questions:**

* Why the surrogate metrics failed to correlate strongly with final PPA results? Given the weak correlation with final PPA metrics, it would be helpful to understand whether alternatives might better reflect the design objectives.
* What were the guideline in developing the benchmark suite? Additional insight into the design process for the benchmark circuits could offer helpful context for other researchers interested in contributing to or building upon this work.

---

### Official Review · Reviewer_WDwZ · 2024-10-31

**Soundness:** 3
**Presentation:** 4
**Contribution:** 3
**Rating:** 5
**Confidence:** 4

**Summary:**

This paper presents two things: a complete framework (ChiPBench) for chip placement with some benchmark circuits and a comparison of different placement algorithms using the framework. The framework is available on GitHub, allowing others to use it as well. The comparison provide some insights on how to use different metrics in AI-based placement algorithms. For instance, MacroHPWL is a commonly used metric in recent works as it is available sooner in the placement process than some other metrics. It turns out, the metric has a rather low correlation with other metric more reflecting of final performance of the placement.

**Strengths:**

The framework seems complete and readily available for others to use.
The analysis using the framework is helpful for those working on new AI-based placement algorithms.

**Weaknesses:**

It is unclear what exact circuit is being used for the "main" results presented in Table 3 and 4. While a reader can obtain details in the appendix for individual circuit results, Table 3 and 4's results should be clearly identified. The "standard cell" is unhelpful (and actually quite confusing: are you placing a single cell?).

**Questions:**

The pitfall of selecting poor intermediate metric is well demonstrated. But in the grand scheme of things, it is unclear how much real-world difference does it make. Take area from Table 5 as an example of the most concrete end metric. Among all the alternatives the difference between the best and worst is 3.8%. Why couldn't we declare the placement problem is largely a solved one?

---

### Official Review · Reviewer_5kac · 2024-11-01

**Soundness:** 3
**Presentation:** 3
**Contribution:** 2
**Rating:** 5
**Confidence:** 5

**Summary:**

The authors propose a new framework, ChipBench, to evaluate the end-to-end performance of AI-based placement algorithms. They also collect a dataset including 20 circuits from various domains and implement EDA flow on these circuits. Finally, they evaluate six placement algorithms in this framework and reveal the weak correlation between the intermediate metrics and the final design PPA.

**Strengths:**

1. The comprehensive summary of current dataset, placement algorithms, and EDA flow.
2. Supported by the open source tool that makes all data accessible to users.
3. The meaningful insights about the weak correlation between the intermediate metrics and the final design PPA.

**Weaknesses:**

1. This framework heavily depends on the OpenROAD framework but the authors don’t compare with it.
2. Section 5 is not essential for the evaluation and the technical novelty is low.
3. The type of PDK is missing in the paper.
4. The analysis merely focuses on the flow but not the inner behavior of placement algorithms.

**Questions:**

I appreciate the authors focusing on the end-to-end performance evaluation of the placement algorithms, which is more concerned by the industry before. From the evaluation results, the weak correlation between the intermediated metrics and the final PPA results are meaningful to researchers.

However, I still have some concerns and suggestions about this work.

1. This framework heavily depends on the OpenROAD framework. I think OpenROAD has provided an easy-to-use EDA toolchain, including core algorithms, framework implementing, tcl scripts, PDK, and sample designs. This work collects more designs from different sources, including OpenROAD. Especially, the authors also cited a project called *OpenROAD flow scripts*, which includes some circuits, support synthesis, and physical design flow, and is more comprehensive than some compared datasets in Table 1. Therefore, more comparison with *OpenROAD flow scripts* could further highlight this work’s unique contributions.
2. Section 5 spends a substantial portion to introduce some placement algorithms, which are just objects to be evaluated not the essentials for the framework. The authors could re-evaluate the content of Section 5, considering integrating it into other relevant sections or removing it entirely.
3. The type of PDK is missing in the paper and this is one of the most basic information for EDA evaluation.
4. In the analysis, the authors provide some observations from the performance of different algorithms, such as how buffer insertion will affect the final TNS and WNS. However, considering the three major types of placement algorithms,  e.g., BBO, gradient-based, and RL, this paper lacks an analysis of the weak correlation and the inner behavior of placement algorithms. Maybe some new evaluation methods should be applied to this task.

---

### Note · Authors · 2024-11-25

I have read and agree with the venue's withdrawal policy on behalf of myself and my co-authors.